# Retrieving the Motion of Beaufort Sea Ice Using Brightness Temperature Data from FY-3D Microwave Radiometer Imager

**DOI:** 10.3390/s22218298

**Published:** 2022-10-29

**Authors:** Kun Ni, Haihua Chen, Lele Li, Xin Meng

**Affiliations:** College of Marine Technology, Faculty of Information Science and Engineering, Ocean University of China, Qingdao 266100, China

**Keywords:** sea ice motion, FengYun-3D, brightness temperature, maximum cross-correlation

## Abstract

Sea ice is an important marine phenomenon in the Arctic region, and it is of great importance to study the motion of Arctic sea ice in the present day when its melting is accelerated by global warming. This study proposes a method to retrieve the motion of sea ice based on the maximum cross-correlation (MCC) and the successive correction method (SCM). The proposed method can apply different scales of search ranges to template matching according to the location of sea ice in the Arctic area. In addition, the data assimilation method can assign different weights to different data. We used 36.5 GHz and 89 GHz brightness temperature (T*_b_*) data from the microwave radiometer imager (MWRI) aboard the Fengyun-3D (FY-3D) satellite, for the first time in the literature, to retrieve the sea ice motion in the Beaufort Sea from January to April 2019. The retrieved sea ice motion results were in good agreement with those obtained from the motion of the buoys. Compared with the data from the buoys, the root mean-squared error (RMSE) of the sea ice motion retrieved from FY-3D/MWRI $T_{b}$ data was 1.1418 cm/s in the zonal direction and 1.0481 cm/s in the meridional direction, and the mean absolute error (MAE) between them was 0.7166 cm/s in the zonal direction and 0.6777 cm/s in the meridional direction. The RMSE between the sea ice motion obtained from the National Snow and Ice Data Center (NSIDC) and the motion of the buoys was 0.9515 cm/s in the zonal direction and 0.67003 cm/s in the meridional direction, and the MAE between them was 0.6576 cm/s in the zonal direction and 0.4922 cm/s in the meridional direction. The RMSE of daily average velocity from the FY-3D/MWRI results and NSIDC data product was 2.2726 cm/s in zonal and 1.9270 cm/s in meridional, and the MAE was 1.5103 cm/s in zonal and 1.1071 cm/s in zonal. The density of the merged data was higher than that obtained from a single polarization or frequency in this paper. The results indicate that FY-3D/MWRI T*_b_* data can retrieve the sea ice motion successfully.

## 1. Introduction

Sea ice in the Arctic region has a strong connection with the global climate, and exercises an important influence on the global water cycle, atmospheric circulation, and sea–air interactions [1,2]. The sea ice motion is an important factor influencing its mass balance [3], and the increase in the velocity of sea ice is among the most important reasons for a reduction in its volume [4]. The Beaufort Sea is an important marginal sea in the Arctic region, which has an important impact on the ecosystem and human activities in the region [5,6]. As the Pacific warm water mass that passes through the Bering Strait to the Arctic Ocean must pass through the Beaufort Strait, changes of thermal conditions in this area can profoundly affect the transport and melt of Arctic sea ice. Therefore, the changes of the sea ice in the Beaufort Sea will also have a deep influence on the transport and melt of the Arctic sea ice [7].

Satellite remote sensing technology has been widely used in research on the Arctic area [7,8,9,10] due to its advantages of good continuity, large spatial and temporal spans, and independence from geographical factors. Passive microwave remote sensing technology can not only overcome the limitations of meteorological conditions such as clouds and sunlight, but also solve the problems of low data volume and high cost in traditional field measurement methods such as buoy observation and ship observation. 

Research into the retrieval of sea ice motion in the Arctic region has been carried out by many scholars. In 1986, Ninnis et al. firstly applied the maximum cross-correlation (MCC) method to retrieve the sea ice motion by using data from the Advanced Very-High-Resolution Radiometer (AVHRR) and obtained the sea ice motion in the eastern Beaufort Sea [11]. Since then, the MCC method has become the dominant means of retrieving the motion of sea ice. In 1998, Kwok et al. retrieved the motion of sea ice by using T*_b_* data from the Special Sensor Microwave Imager (SSM/I), and compared the results with the motion of buoys, the motion retrieved from Synthetic Aperture Radar (SAR) [12]. In 2000, Martin et al. used 85.5 GHz data from the SSM/I to retrieve the sea ice motion in the Arctic area and obtained accurate results of retrieval in comparison with the motion of buoys [13]. Methods of data assimilation have also been widely used to retrieve the sea ice motion. The product released by the NSIDC combines wind and buoy data with the optimal interpolation method to obtain the measurement of sea ice motion [14]. The product for the measurement of sea ice motion released by the French Research Institute for Exploitation of the Sea (Ifremer) also combines the sea ice motion from the Quick Scatterometer (QuikSCAT) and SSM/I [15]. Many improvements have been made to the MCC method as well; Lavergne et al. applied the bilinear interpolation to the MCC method to reduce the quantization noise caused by MCC in the retrieval of the sea ice motion [16], and Ezaty et al. applied Laplacian computation to preprocess the data and merge motion vectors obtained from different polarizations [17]. Liu et al. applied wavelets to highlight the features of sea ice and improved the accuracy of the retrieved sea ice motion [18]. Kwok et al. used SAR data to retrieve the field of sea ice motion in the Alaskan region by using an algorithm that combines pattern matching and feature matching in 1990 [19]. In 2017, Muckenhuber et al. released an open-source sea ice drift algorithm which combined the feature tracking and pattern matching methods to retrieve the motion of sea ice using SAR data [20]. The same method was used by Wang et al. in 2019 to retrieve the sea ice motion using SAR data, which obtained good results [21]. In 2017, Wang et al. applied the Laplacian of Gaussian (LOG) filter, which can smoothen the noises as well as strengthen the edges of the features in data, to the retrieval of the motion of sea ice and successfully obtained the motion from HY-2 scanning microwave radiometer (RM) T*_b_* data [22]. 

In terms of the quality of different data products to measure the motion of sea ice, scholars have also carried out a great deal of research. In 2020, Shi et al. assessed the sea ice motion products based on the International Arctic Buoy Programme (IABP) ice drift buoys [23]. Wang et al. used buoy data from the Multidisciplinary Drifting Observatory for the Study of Arctic Climate (MOSAiC) to validate the quality of sea ice motion data products in 2021 [24]. 

The Fengyun-3D (FY-3D) meteorological satellite is China’s second-generation polar-orbiting meteorological satellite that was launched in 2017. It is a part of the Global Earth Observation System (GEOSS). Data from FY-3D have been widely used in the research of the Arctic region; the data products of sea ice concentration (SIC) and thickness of snow based on the data from FY-3D have already been released and widely used [25,26,27,28]. However, the data from FY-3D has not been used to retrieve the sea ice motion in the Arctic region. In order to have a better knowledge of the motion of the sea ice and expand the application of the T*_b_* data from FY-3D/MWRI, we retrieved the sea ice motion using the T*_b_* data from FY-3D/MWRI. 

In this study, we proposed a method to retrieve the sea ice motion based on the MCC and SCM methods. We used T*_b_* data from the FY-3D/MWRI to determine the sea ice motion for the first time in the literature, to the best of our knowledge. The results were in good agreement with the motion obtained from buoys, and incurred an error similar to that obtained by the data product released by the NSIDC. 

The remainder of this paper is organized as follows: Section 2 introduces the data and methods used in this study, including methods of data preprocessing, the proposed algorithm to retrieve the sea ice motion, and the algorithm used for data assimilation. Section 3 presents the results and validates the proposed method. We further discuss the results and limitations of the proposed method in Section 4, and provide the conclusions of this study in Section 5.

## 2. Materials and Methods

### 2.1. Data

The data used in this study include FY-3D/MWRI T*_b_* data acquired from the National Satellite Meteorological Center (NSMC), SIC data retrieved from FY-3D/MWRI T*_b_* data released by the Key Lab of Polar Oceanography and Global Ocean Changes (POGOC) [25], the sea ice motion product provided by the NSIDC [29], buoy data from the IABP, and wind velocity data from the National Centers for Environmental Prediction and National Center for Atmospheric Research (NCEP/NCAR). The basic information of the data used in this study is shown in following Table 1.

#### 2.1.1. FY-3D/MWRI T_b_ Data

FY-3D carries a Microwave Radiometer Imager with 5 frequencies of 10.65, 18.7, 23.8, 36.5, and 89 GHz, and the data of each frequency consists of two types of polarization: horizontal and vertical. The information about the data is shown in Table 2. 

We used 36.5 GHz and 89 GHz data to retrieve the sea ice motion in the Beaufort Sea. The 89 GHz data have a high resolution but are significantly affected by the atmospheric conditions. Compared with these data, the 36.5 GHz data have a low resolution but are less affected by the atmospheric conditions. We chose data at these two frequencies to combine their advantages. To improve the accuracy of the results of retrieval, the T*_b_* data used in this study were cross-calibrated with the T*_b_* data from the Advanced Microwave Scanning Radiometer 2 (AMSR2) using the calibration method created by Tang et al. [30].

#### 2.1.2. Sea Ice Concentration Data

Sea ice concentration greater than 15% is the normal state of an ice-covered region [14]. In this study, we used the sea ice concentration data retrieved from FY-3D/MWRI T*_b_* data to mark out the region with a sea ice concentration lower than 15% in the T*_b_* data. The SIC data can be downloaded from the website of the Key Lab of Polar Oceanography and Global Ocean Changes (POGOC) (http://coas.ouc.edu.cn/pogoc/2018/1220/c15472a231971/page.htm (accessed on 22 April 2021)).

#### 2.1.3. Buoy Data

IABP is a project supported by multiple international agencies which deploys and maintains the buoys in the Pacific Arctic region for the purpose of collecting oceanic and meteorological data. These buoys report their latitudinal and longitudinal positions at 12:00 and 24:00 (GMT time) each day. We identified the same buoys on the starting day and the day after a 3-day interval, calculated their velocities at 12:00 and 24:00 based on their positions, and then the two velocities were averaged to provide the velocity of each buoy. The interval of the velocity was three days.

#### 2.1.4. NSIDC Sea Ice Motion Product

The product for the sea ice motion released by the NSIDC [29] is an international daily product with a spatial resolution of 25 km. This product combines the velocity of the wind, motion of buoys and sea ice motion derived from satellite instruments by the optimal interpolation method. We used this data product to further validate the sea ice motion retrieved from the FY-3D/MWRI.

#### 2.1.5. NCEP/NCAR Wind Data

Considering the effect of wind velocity on sea ice motion [31,32,33], we assumed that sea ice moves in the geostrophic wind direction with a magnitude of 1% [14]. We merged the NCEP/NCAR reanalysis wind velocity data with the sea ice motion retrieved from FY-3D/MWRI T*_b_* data and the motion of the buoys as the final result of the sea ice motion [14]. The wind data we used in this study can be downloaded from the website of the NOAA (https://psl.noaa.gov/data/gridded/data.ncep.reanalysis.html) accessed on 10 April 2021.

### 2.2. Data Preprocessing

Firstly, we mapped all kinds of data used in this study onto 12.5 km Equal-Area Scalable Earth (EASE) Grids. This projection coordinate system is centered on the North Pole, with the positive direction of the X-axis ranging from 90 degrees west to 90 degrees east, and the positive direction of the Y-axis ranging from 0 degrees east to 180 degrees east. 

The FY-3D/MWRI T*_b_* data was separated into data received in the ascending orbit and those obtained in the descending orbit. The interval of each data item was about one hour. For each grid cell, the FY-3D/MWRI T*_b_* data observed over 24 h were combined to obtain the average daily T*_b_*. We then used the SIC data to mark out the water region, and chose grid cells with SIC > 15% as the region covered by sea ice. The FY-3D/MWRI T*_b_* data with SIC < 15% was deleted.

To highlight the features of sea ice in the FY-3D/MWRI T*_b_* images, the LOG filter was applied to the FY-3D/MWRI T*_b_* data before retrieving the sea ice motion from them. We firstly smoothed the images by using the Gaussian operator to suppress noise, and then strengthened their edges by calculating the zero intersection of their second-order Laplaction derivatives [22]. We used f(x,y) to represent the image and h(x,y) to represent the image after it had been subjected to LOG filtering:(1)$$h\left(x,y\right)=\nabla^{2}\left(G\left(x,y\right)⊗f\left(x,y\right)\right)$$
where G(X,Y) is the Gaussian filter and $\nabla^{2}$ is the Laplacian calculation [22]. 

### 2.3. Method to Retrieve Sea Ice Motion

In this study, we used two FY-3D/MWRI T*_b_* data images with a 3-day interval to retrieve the sea ice motion. This interval ensured both a high correlation between images and the retrieval of the sea ice motion at relatively low velocities [13]. During this interval, the change of sea ice was not too severe to be retrieved but the interval was also enough for most sea ice to move more than one pixel in the images. 

Figure 1 shows the process of the whole work. After preprocessing the data used in this study, we applied the FY-3D/MWRI T*_b_* data to the improved MCC method and obtained the preliminary result of the sea ice motion, and then used the buoy data and wind data to revise the sea ice motion by the data assimilation method based on SCM. Then, we validated the sea ice motion retrieved using the buoy data and NSIDC data product. We discuss the methods used in this study in detail in the following three sections. 

#### 2.3.1. Improved MCC Method for Obtaining the Sea Ice Motion

The MCC method is a template-matching method based on the cross-correlation coefficients between two image. Figure 2 shows the process of the MCC method, and the details of the MCC method have been provided by Ninnis [11].

In this study, we selected a template of size 7 × 7 pixels from the image of the starting day, then traversed the search area of image after a 3-day interval with the selected template, and calculated the correlation coefficient between the template and the search area. The size of the template was obtained after extensive experiments, which contained enough information on sea ice characteristics while maintaining a high computation speed.

After experiments and analysis, we set the threshold of the cross-correlation coefficient to 0.4; this threshold ensured that the results of retrieval could correctly reflect the sea ice motion and reduced the outliers in the results as well. If the maximum cross-correlation coefficient was unique and greater than the threshold, its location was assumed to be that to which the sea ice had moved after three days. We multiplied the number of moving grids by the resolution to obtain the distance moved by the sea ice, and then divided it by the interval of three days to obtain its velocity. If the maximum cross-correlation coefficient was lower than the threshold or was not unique, the grid was not used to compute the velocity of sea ice.

One flaw in the MCC method is the quantization error owing to the low resolution of data that caused the field of motion to appear quantized, with a poor angular distribution [16]. We addressed this flaw by performing a linear interpolation to oversample the matrix of the cross-correlation coefficients. 

The search area has a significant impact on the results of matching in the template-matching process, and its size determines the maximum velocity of sea ice motion that can be retrieved. However, the sea ice motion in the Arctic region is complex. In light of the differences in the motions of different regions, we used the inverse distance weighted (IDW) method, a data interpolation method based on the Euclidean distance among data points, to calculate the size of the search area. The method can be described by the following two equations:(2)$$\lambda_{i}=\frac{\;\frac{1}{d_{i}}}{\left(\sum_{i=1}^{n}\frac{1}{d_{i}}\right)}$$
(3)$$\hat{Z}\left(X_{0},Y_{0}\right)=\overset{n}{\underset{i=1}{\sum }}\lambda_{i}Z\left(X_{i},Y_{j}\right)$$

$Z\left(X_{i},Y_{j}\right)$ in the above is the value of the known grid, $\hat{Z}\left(X_{0},Y_{0}\right)$ is the value of an unknown grid calculated by the IDW method, $d_{i}$ is the distance between the known and unknown grids, and *n* is the number of known grids. We used the IDW method to interpolate the velocity of the unknown grids based on the data of buoys within 417 km of it. The known data points were given different weights by Equation (2), and then the weights $\lambda_{i}$ and the known points were used to interpolate the value of an unknown point. The results of interpolation were used to calculate the scope of the search area in the MCC method. If there was no buoy around the unknown grid, we used the maximum velocity of buoys in the Arctic region to calculate the scope of the search area. 

The sea ice motion has a high consistency within a range of 35 × 35 grids [22]. After obtaining the results of retrieval, we removed grids with differences greater than three times the standard deviation from the mean in the range of 35 × 35 grids. We then merged the velocity of wind and the motion of buoys into the retrieved velocity of sea ice by the data assimilation method based on the SCM to determine the sea ice motion. The data assimilation method used in this study is discussed in Section 2.3.2. 

#### 2.3.2. Data Assimilation

In this study, we used the SCM, an empirically based analysis method, to perform data assimilation. The SCM was developed by the Swedish scientists Bergthorsson, Doos [34,35] and Cressman [34,36] at the National Weather Service (NWS) of the USA. It can simultaneously consider the effects of distance and different sources of data, and is simple enough and sufficiently efficient to provide a reasonable analysis. The method can be described as follows:

We assume the initial data $f^{0}$ as the background data $f^{b}$, where $f_{i}^{0}=f_{i}^{b}$. $f_{i}^{b}$ is the value of the background data on grid *I*, $f_{i}^{0}$ is the zeroth iterative estimates of grid *i*, and the final result is obtained by iteratively computing the data *n* times. The formula is as follows [34]:(4)$$f_{i}^{n+1}=f_{i}^{n}+\frac{\sum_{k=1}^{K_{i}^{n}}w_{ik}^{n}\left(f_{k}^{o}-f_{k}^{n}\right)}{\sum_{k=1}^{K_{i}^{n}}w_{ik}^{n}+\varepsilon^{2}}$$
(5)$$\begin{array}{c} w_{ik}^{n}=\frac{R_{n}^{2}-r_{ik}^{2}}{R_{n}^{2}+r_{ik}^{2}}\;\left(r_{ik}^{2}\leq R_{n}^{2}\right)\\ w_{ik}^{n}=0\;\;\left(r_{ik}^{2}>R_{n}^{2}\right) \end{array}\}$$

The weights $w_{ik}^{n}$ in this study were determined as shown above. $r_{ik}$ is the distance between the observation grids and the interpolated grids, and $R_{n}$ is the scope of influence of the observational data. 

$f_{i}^{n}$ is the value of the estimate of the *n*th iteration on grid point *i*, $f_{k}^{o}$ is the observational data item on grid *k*, $f_{k}^{n}$ represents the estimates of grid *k*, $\varepsilon^{2}$ is the estimated ratio of the variance of the observational error to that of the background error, and $K_{i}^{n}$ is the total number of observations within distance $R_{n}$ from the interpolated grid point. 

In light of the effects of wind on the sea ice motion [33], we assigned different weights to the sea ice motion after data assimilation, wind data, and buoy data. The weights of different sources are of the form of Equation (6) [14]:(6)$$w=Ce\left(-\frac{d}{D}\right)$$

w is weight, and C is a source-based coefficient determined by the correlation coefficient between different sources and buoys. As we set the motion of buoys as the true value of the sea ice motion, the value of C of buoy motion was one, and the value of C of the sea ice motion retrieved by the satellite and that of the wind were their correlation coefficients with the buoys. *d* is the Euclidean distance between the pixel in question and the motion estimate on the EASE-Grid, and *D* is the length scale (constant) over which the estimates are correlated. *D* was set to a value of 417 km, referring to the setting of NSIDC [14]. Using Equation (6) we can calculate the value of the weight w for every source from every day. 

The final result of sea ice motion was obtained by weighting the sum of these three kinds of data, as shown in Equation (7):(7)$$SIV=w_{w}v_{w}+w_{b}v_{b}+w_{sat}v_{sat}$$
where *SIV* is the final velocity of sea ice, and $w_{w}$, $w_{b}$ and $w_{sat}$ are the weights assigned to the velocities of wind, buoys and sea ice retrieved from the satellite, respectively. We used the 15 sea ice motion vectors with the largest weights with in a 417 km radius of each grid’s cell to calculate the final value of *SIV* [14].

#### 2.3.3. Quality Assessment

We used MAE (δ) and RMSE (σ) to validate the data and assess the accuracy of determining the sea ice motion. δ and σ were calculated as follows:

The δ: (8)$$\delta =\frac{1}{n}\overset{n}{\underset{i=1}{\sum }}\left|SIV_{i}-V_{BUOY_{i}}\right|$$

The  σ:(9)$$\sigma =\frac{1}{n}\sqrt{\overset{n}{\underset{i=1}{\sum }}{\left(SIV_{i}-V_{BUOY_{i}}\right)}^{2}}$$

In the above, SIV is the velocity of sea ice motion and $V_{BUOY}$ is the velocity of the buoy. We compared the velocities in two directions: the zonal direction (parallel to the latitude), and the meridional direction (parallel to the longitude). We used $\delta_{Z}$ and $\sigma_{Z}$ to represent the δ and σ in the zonal direction, respectively, and $\delta_{M}$ and $\sigma_{M}$ to represent them in the meridional direction, respectively. 

## 3. Results

### 3.1. Determining the Sea Ice Motion by Using the Improved MCC Method

We selected the area of the Beaufort Sea between 120° W and 160° W and south of 80° N to retrieve the field of sea ice motion. This area is shown in Figure 3.

We applied our improved method to the FY-3D/MWRI T*_b_* data to obtain the field of sea ice motion in the Beaufort Sea from January to April of 2019 at different frequencies and polarizations.

Table 3 and Table 4 show the δ and σ in the retrieved sea ice motion, and Figure 4 and Figure 5 show the schematic of the retrieved sea ice motion. Figure 6 and Figure 7 show comparisons between the retrieved sea ice motion and the motion of the buoys from January 2019 to April 2019 in Beaufort Sea.

Figure 4 and Figure 5 show the sea ice motion on the first day of each month retrieved by FY-3D/MWRI T*_b_* data. The red vectors represent the sea ice motion vectors retrieved from FY-3D/MWRI T*_b_* data and the black vectors represent the motion of the buoys. The filled color in the figures represents the value of sea ice velocity. The sea ice motion retrieved using the MCC method had a number of missing data items, and the directions of the buoys were not very consistent with the directions of the sea ice motion retrieved from FY-3D. 

We compared the values of the sea ice velocity in the zonal and meridional directions, respectively, and the results are shown in Figure 6 and Figure 7, respectively. The scatter plot shows that although the distribution of the data was lightly fragmented, they were roughly distributed around the red dotted line. This distribution shows that the sea ice motion retrieved from FY-3D can roughly reflect the true state of sea ice motion in the Arctic region. The σ of retrieved sea ice motion ranged from 2.5713 cm/s to 5.3117 cm/s and the δ of retrieved sea ice motion ranged from 1.8389 cm/s to 4.0861 cm/s. The results show that directly applying the MCC method to passive microwave data cannot yield very accurate results of sea ice motion. The research of NSIDC also yielded errors in the sea ice motion retrieved from passive microwave data by using the MCC method. The σ of the sea ice motion retrieved from SSM/I on different components were, respectively, 4.16 cm/s and 4.23 cm/s [29].

To reduce the number of missing data in the results of retrieval and improve the accuracy of the results of retrieval, we combined these results into one field of sea ice motion. The process as follows:(10)$$SIV\{\begin{array}{c} \frac{H_{SIV}+V_{SIV}}{2}\left(V_{SIV}\neq nan,H_{SIV}\neq nan\right)\\ H_{SIV}\;\left(H_{SIV}\neq nan\right)\\ V_{SIV}\;\left(V_{SIV}\neq nan\right) \end{array}$$

*SIV* is the sea ice velocity, and $H_{SIV}$ and $V_{SIV}$ are the values of *SIV* retrieved from the *H* and *V* polarizations, respectively. For grids with $H_{SIV}$ and $V_{SIV}$, we calculated the average of two results as the final result. For the grids containing only $V_{SIV}$ or $H_{SIV}$, we chose the non-null value.

After combining the results from the *H* and *V* polarizations, we used the following formula to combine the results of retrieval obtained at two frequencies:(11)$$SIV\{\begin{array}{c} SIV_{36.5}\;\left(SIV_{36.5}\neq nan,SIV_{89}=nan\right)\\ SIV_{89}\;\left(SIV_{89}\neq nan,SIV_{36.5}=nan\right)\\ SIV_{36.5}\;\left(SIV_{89}=nan,SIV_{36.5}\neq nan\right) \end{array}$$

$SIV_{36.5}$ and $SIV_{89}$ are the sea ice velocities retrieved at 36.5 GHz and 89 GHz, respectively. As the sea ice motion based on 36.5 GHz T*_b_* data had smaller values of σ and δ than those based on 89 GHz T*_b_* data, we used the sea ice motion retrieved from 36.5 GHz to fill the vacant grids in the sea ice motion field. 

After combining the results of retrieval at different polarizations and frequencies, the data on δ and σ changed by very little (Table 5, Figure 8) but the field of sea ice motion was rendered denser (Figure 9). The results still contained large errors compared with the buoy motion data. 

### 3.2. Assimilating Data on the Sea Ice Motion 

To further improve the accuracy of the retrieved sea ice motion, we used the SCM to assimilate the buoy motion data with it. Before assimilation, we filled the vacant grids in the sea ice motion data by using the IDW method. The resulting motion was used as the initial background field, and the buoy motion data was used as observational data for the SCM. $R_{0}$ in Equation (5) was set to 417 km, an empirical value recommended by the NSIDC [14]. $\varepsilon^{2}$ in Equation (4) was set to zero as the motion of the buoy representing true value. Then, we merged the wind data, buoy data, and sea ice motion retrieved from FY-3D/MWRI T*_b_* into the field of sea ice motion using Equations (6) and (7). 

Because of the small amount of buoys, this study uses buoy motion data for data assimilation as well as the same data for comparison and validation of the results of retrieval.

To verify the accuracy of our data assimilation method, we randomly selected two-thirds of the daily buoy motion data for assimilation and used the remaining one-third to test the results of data assimilation. We matched the remaining one-third buoy data with the sea ice motion from January to April of 2019, obtaining about 500 matching data points. Figure 10 shows that when using two-thirds of buoy motion data for data assimilation, the results of the sea ice motion were accurate compared with the motion of the remaining one-third buoys. σ in zonal and meridional were 1.9688 cm/s and 1.9202 cm/s, respectively, and δ in zonal and meridional were 1.2667 cm/s and 1.3007 cm/s, respectively. The vectors of sea ice motion have good agreements with the buoy motion data that have not been used in data assimilation. The result shows that the data assimilation method can be used to improve the quality of the sea ice motion data.

We applied all the buoy motion data for data assimilation to obtain the sea ice motion in the Beaufort Sea. Figure 11a,b show that the result delivered a high accuracy. σ in zonal and δ in zonal were, respectively, 1.1418 cm/s and 0.7166 cm/s, and the values of σ and δ in meridional were, respectively, 1.0481 cm/s and 0.6777 cm/s. 

In Figure 4, Figure 5, Figure 9, Figure 12 and Figure 13, the red vectors represent the direction of sea ice motion and the black vectors represent the directions of the buoys. The sea ice motion vectors in 7 × 7 grids were averaged to make the map clearer. 

Comparing Figure 12 and Figure 9, the field of sea ice motion retrieved from FY-3D/MWRI T*_b_* data shown in Figure 12 is denser. In Figure 12, the sea ice motion retrieved from FY-3D/MWRI T*_b_* data and NSIDC data product all have good agreements with the buoy motion. Though there are differences existing in the NSIDC data product and sea ice motion retrieved from FY-3D/MWRI T*_b_* data, they can all reflect the true state of sea ice motion in the Arctic region.

By calculating the monthly average velocity of sea ice motion from Jan 2019 to April 2019, we can compare the long-time general trend of sea ice motion retrieved from FY-3D/MWRI T*_b_* data and from the NSIDC data product. The general trends of sea ice motion retrieved from FY-3D/MWRI T*_b_* data and the NSIDC data product are similar to each other. In a nearshore sea region, we may underestimate the velocity of sea ice motion compared with the sea ice motion from the NSIDC data product. 

### 3.3. Comparison with Data from the NSIDC Product

In this section, we compare the sea ice motion obtained from the NSIDC and the sea ice motion retrieved from FY-3D/MWRI T*_b_* data.

The “Polar Pathfinder Daily 25 km EASE-Grid Sea Ice Motion Vectors” dataset [29], released by the NSIDC, is an international mainstream sea ice motion that has been widely used for numerical validation and data assimilation in research [24]. This dataset combines the motion of the buoy, the reanalyzed wide velocity, and the sea ice motion retrieved by the satellite to yield gridded fields of the motions of sea ice. We averaged the NSIDC daily sea ice motion for three days, and considered the averaged result as sea ice motion with a time separation of 3 days. Then, we compared the averaged sea ice motion with the retrieved sea ice motion from FY-3D/ MWRI T*_b_* data. 

Figure 11 shows that the sea ice motion retrieved from FY-3D/MWRI T*_b_* data and the sea ice motion from NSIDC data product all had good agreements with the buoy data. Comparing Table 5 and Table 6, the errors of the sea ice motion retrieved from FY-3D/MWRI T*_b_* become very small, though they are bigger than the errors of the data product from NSIDC. The comparison shows that both can reflect the sea ice motion in the Arctic region, and the data assimilation method can improve the accuracy of the sea ice motion result.

Figure 12 shows that, though the sea ice motion in some areas is different between the sea ice motion retrieved from FY-3D/MWRI T*_b_* data and the data product from NSIDC, the directions of the sea ice motion vectors retrieved from FY-3D/MWRI T*_b_* data and NSIDC data product are consistent with the sea ice motion vectors of buoys. 

We also computed the daily average velocity of the sea ice motion retrieved from different sources. The comparisons of daily average velocity are shown in Figure 14a–d from January to April 2019. 

We can see from Figure 14a,b that sea ice motion retrieved from FY-3D/MWRI T*_b_* data matched not very well with NSIDC data product. By statistical calculation, the σ of daily average velocity from FY-3D/MWRI results and NSIDC product is 3.1759 cm/s in zonal and 3.3277 cm/s in meridional, and the δ is 2.5738 cm/s in zonal and 2.6673 cm/s in meridional, respectively. 

Figure 14c,d show that after data assimilation, the sea ice motion retrieved from FY-3D/MWRI T*_b_* data matched well with the NSIDC data product. The σ of daily velocity from the FY-3D/MWRI results and the NSIDC product is 2.2726 cm/s in zonal and 1.9270 cm/s in meridional, and the δ is 1.5103 cm/s in zonal and 1.1071 cm/s in meridional direction. 

The sea ice motion retrieved from the FY-3D/MWRI T*_b_* data shows that the FY-3D/MWRI T*_b_* data can be used to retrieve the sea ice motion in the Arctic region, and that the data quality has been improved greatly by the improved MCC method and data assimilation method. By comparing the daily average sea ice motion retrieved from FY-3D/MWRI and the NSIDC product, we further illustrated the accuracy of the results of retrieval.

## 4. Discussion

Comparing Table 6 and Table 5, we can find that the quality of the data was improved considerably by merging the data. The σ in zonal $\sigma_{z}$ decreased from 3.6159 cm/s to 1.1418 cm/s, the δ in zonal $\delta_{z}$ decreased from 2.6824 cm/s to 0.7165 cm/s, σ in meridional decreased from 4.4857 cm/s to 1.0481 cm/s, and δ in meridional decreased from 3.2437 cm/s to 0.6777 cm/s. The comparison of Table 6 and Table 5 show that the SCM significantly improved the quality of the product in terms of the sea ice motion. Because the data-merging algorithm also considered the influence of neighboring motions of sea ice, its results were smoother than those of the improved MCC method. 

As shown in Figure 4, Figure 5, Figure 6 and Figure 7 and Table 3 and Table 4, the results of the sea ice motion obtained by the improved MCC method contained many vacant grids and errors. Figure 8 and Figure 9 and Table 5 show that the density of the data were improved by merging information at different polarizations and frequencies, though errors persisted. The reason of the error may be the stable threshold we set in the MCC method. The sea ice motion in the Beaufort Sea is complex and variables, the stable threshold may lead to wrong matches in the results of retrieval. It thus might be useful to set different thresholds for different parts of the sea in future work. 

Compared with the NSIDC’s data product, our product underestimated velocity in the nearshore sea, as can be seen in Figure 12 and Figure 13. Regardless of the month, of the four considered in this study, the sea ice motion retrieved using FY-3D/MWRI T*_b_* data nearshore was always slower than that of the NSIDC product (Figure 13). The three-day sea ice motion (Figure 12) also shows this underestimation. This might have occurred because the range of search nearshore was too small. We used buoy motion data to compute the range of the search area, but the small number of buoys nearshore might have led to an underestimation of the search scope in this area and thus the velocity of sea ice. The SCM is an empirical algorithm, and its result is thus greatly influenced by distance. Therefore, grids far from the measured data are more likely to be poorly estimated. To improve the accuracy of the product in terms of the sea ice motion, we need to use a larger search area and further improve the means of data assimilation. 

## 5. Conclusions

In this study, we improved the MCC method by applying the IDW method, Gaussian of Laplacian filtering, and linear interpolation to retrieve the sea ice motion. In addition, we developed a data assimilation algorithm based on the SCM that could merge different kinds of data. We applied it to the FY-3D/MWRI T*_b_* data to retrieve the sea ice motion in the Beaufort Sea from January to April 2019, for the first time in the literature to the best of our knowledge. We validated our product with data from buoys deployed by the IABP and the data product from NSIDC; the results verified its accuracy. However, compared with the buoy data, error in the sea ice motion retrieved by using FY-3D/MWRI T*_b_* data ($\delta_{Z}$ = 0.7166 cm/s,$\;\delta_{M}$ = 0.6777 cm/s,$\;\sigma_{Z}$ = 1.1418 cm/s,$\;\sigma_{M}$ = 1.0481 cm/s) was larger than the error in the NSIDC product for the same phenomenon ($\delta_{Z}$ = 0.6576 cm/s,$\;\delta_{M}$ = 0.4922 cm/s,$\;\sigma_{Z}$ = 0.9515 cm/s,$\;\sigma_{M}$ = 0.6700 cm/s). In the nearshore sea, our product might have underestimated the velocity of sea ice compared with the data product from NSIDC, as shown in Figure 12 and Figure 13. Figure 14 shows that the daily velocity of sea ice retrieved using FY-3D/MWRI T*_b_* data have many unique values compared with the data product from the NSIDC, and this shortcoming needs to be remedied. In future studies, we plan to improve the method by using variable correlation thresholds according to the region, and further improving the method of data assimilation to better estimate the sea ice motion in region with few buoys. 

## Figures and Tables

**Figure 1 sensors-22-08298-f001:**
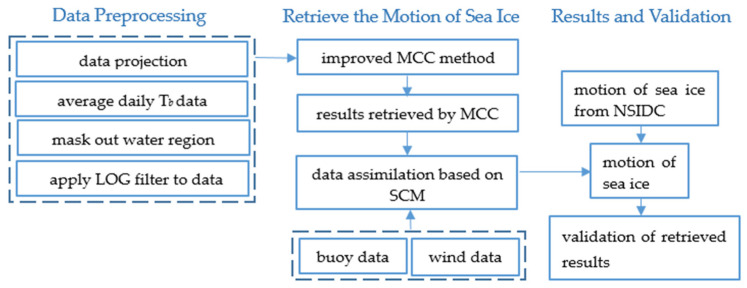
Flowchart of retrieving the sea ice motion and validation.

**Figure 2 sensors-22-08298-f002:**
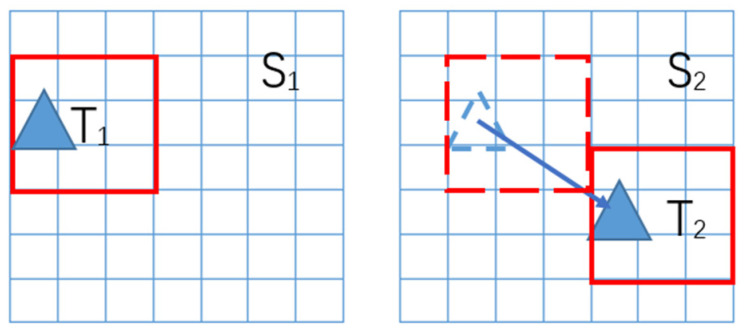
Pattern-matching process. T_1_ is the template from the S_1_ and T_2_ is the matching position of T_1_ in S_2_. Select a template T_1_ from S_1_, then traverse the S_2_ with T_1_, and calculate the correlation coefficients between S_2_ and T_1_, T_2_ is the position with highest value of coefficient.

**Figure 3 sensors-22-08298-f003:**
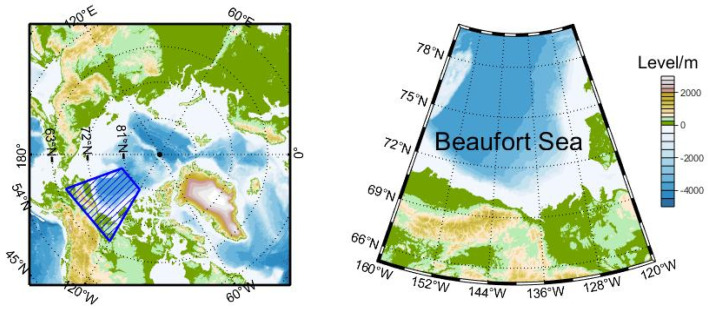
Research area.

**Figure 4 sensors-22-08298-f004:**
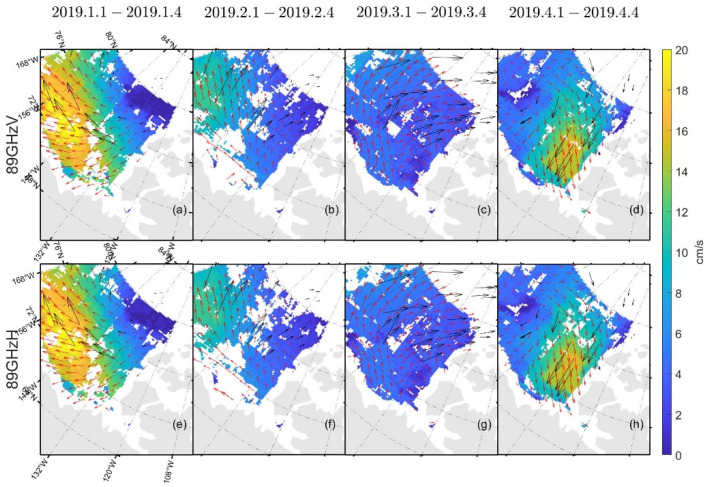
Schematic diagram of the movement of sea ice retrieved by FY-3D/MWRI T*_b_* data at 89 GHz. Figure (**a**–**d**) show sea ice motion retrieved from FY-3D/MWRI T*_b_* data at 89 GHz with V polarization. Figure (**e**–**h**) show sea ice motion retrieved from FY-3D/MWRI T*_b_* data at 89 GHz with H polarization.

**Figure 5 sensors-22-08298-f005:**
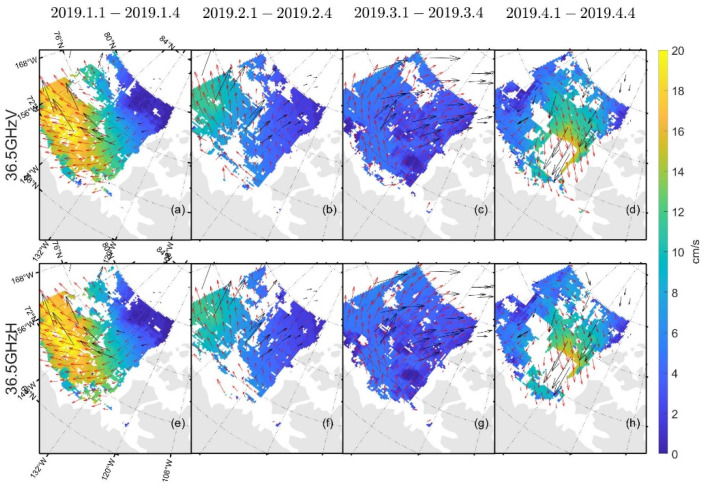
Schematic diagram of the movement of sea ice retrieved by FY-3D/MWRI T*_b_* data at 36.5 GHz. Figure (**a**–**d**) show sea ice motion retrieved from FY-3D/MWRI T*_b_* data at 36.5 GHz with V polarization. Figure (**e**–**h**) show sea ice motion retrieved from FY-3D/MWRI T*_b_* data at 36.5 GHz with H polarization.

**Figure 6 sensors-22-08298-f006:**
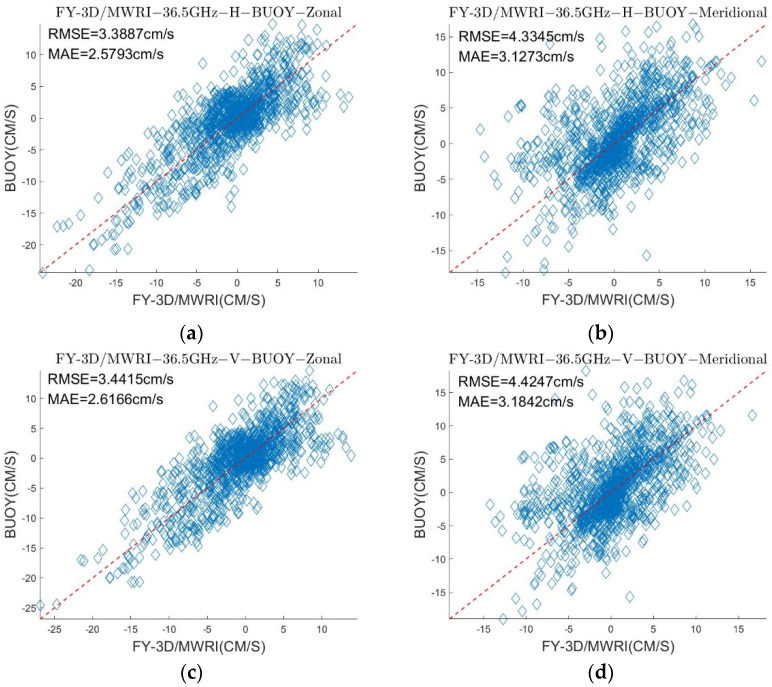
Comparison between the sea ice motion, retrieved from FY-3D/MWRI T*_b_* data at 36.5 GHz with different polarizations, and the motion of the buoy. (**a**) is a scatter diagram of the sea ice motion retrieved by using FY-3D/MWRI T*_b_* data at 36.5 GHz with H polarization and the motion of the buoy in the zonal direction, (**b**) is a scatter diagram of the sea ice motion retrieved by using FY-3D/MWRI T*_b_* data at 36.5 GHz with H polarization and the motion of the buoy in the meridional direction, (**c**) is a scatter diagram of the sea ice motion retrieved by using FY-3D/MWRI T*_b_* data at 36.5 GHz with V polarization and the motion of the buoy in the zonal direction, and (**d**) is a scatter diagram of the sea ice motion retrieved by using FY-3D/MWRI T*_b_* data at 36.5 GHz with V polarization and the motion of the buoy in the meridional direction.

**Figure 7 sensors-22-08298-f007:**
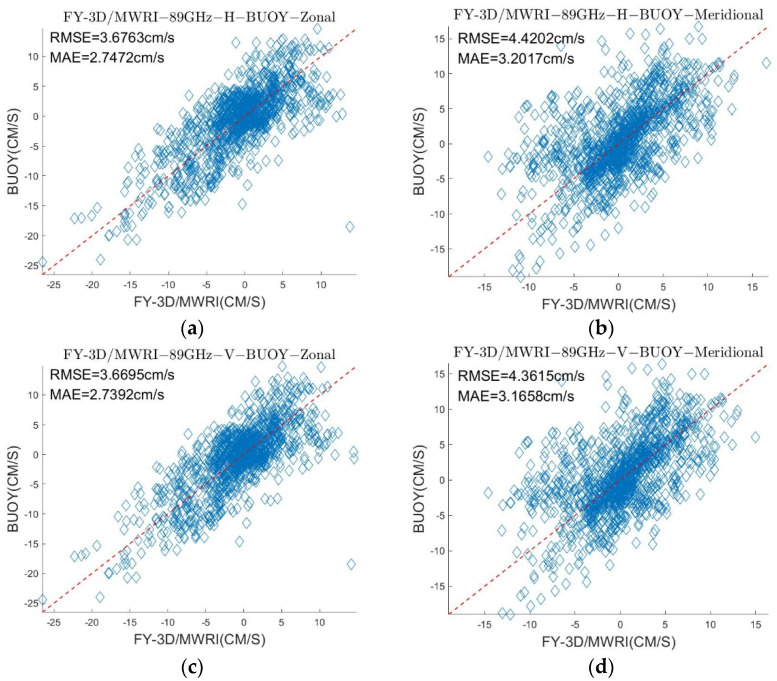
Comparison between the sea ice motion, retrieved from FY-3D/MWRI T*_b_* data at 89 GHz with different polarizations, and the motion of the buoy. (**a**) is a scatter diagram of the sea ice motion retrieved by using FY-3D/MWRI T*_b_* data at 89 GHz with H polarization and the motion of the buoy in the zonal direction, (**b**) is a scatter diagram of the sea ice motion retrieved by using FY-3D/MWRI T*_b_* data at 89 GHz with H polarization and the motion of the buoy in the meridional direction, (**c**) is a scatter diagram of the sea ice motion retrieved by using FY-3D/MWRI T*_b_* data at 89 GHz with V polarization and the motion of the buoy in the zonal direction, and (**d**) is a scatter diagram of the sea ice motion retrieved by using FY-3D/MWRI T*_b_* data at 89 GHz with V polarization and the motion of the buoy in the meridional direction.

**Figure 8 sensors-22-08298-f008:**
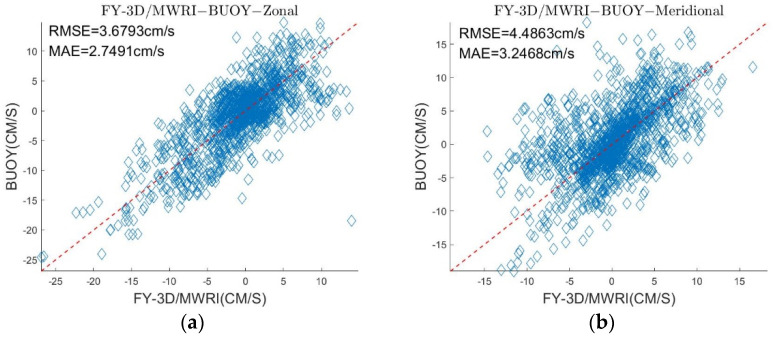
(**a**) is a comparison of the motions of sea ice retrieved by FY-3D/MWRI T*_b_* data merging different polarizations and frequencies with the motion of the buoy in the zonal direction, (**b**) is a comparison of the motions of sea ice retrieved by FY-3D/MWRI T*_b_* data merging different polarizations and frequencies with the motion of the buoy in the meridional direction.

**Figure 9 sensors-22-08298-f009:**
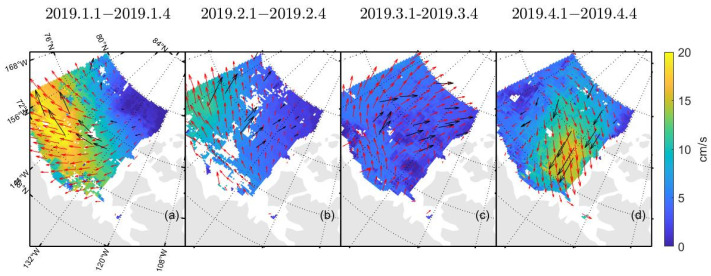
The sea ice motion obtained by merging different frequencies and polarizations. (**a**) is the sea ice motion on 1 January 2019, (**b**) is the sea ice motion on 1 February 2019, (**c**) is the sea ice motion on 1 March 2019, (**d**) is the sea ice motion on 1 April 2019.

**Figure 10 sensors-22-08298-f010:**
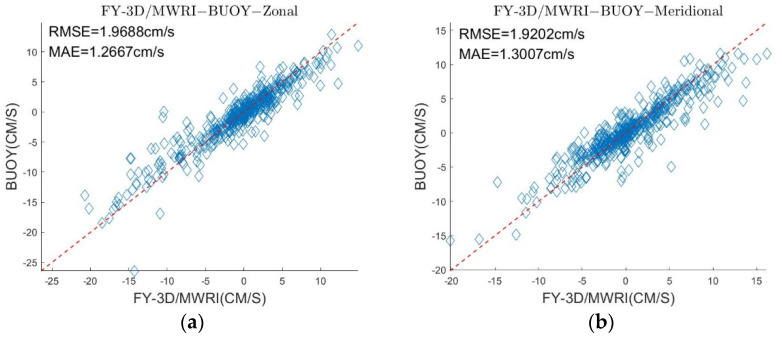
(**a**) is a comparison of the retrieved sea ice motion and one-third of the buoy motion data in the zonal direction. (**b**) is a comparison of the retrieved sea ice motion and one-third of the buoy motion data in the meridional direction.

**Figure 11 sensors-22-08298-f011:**
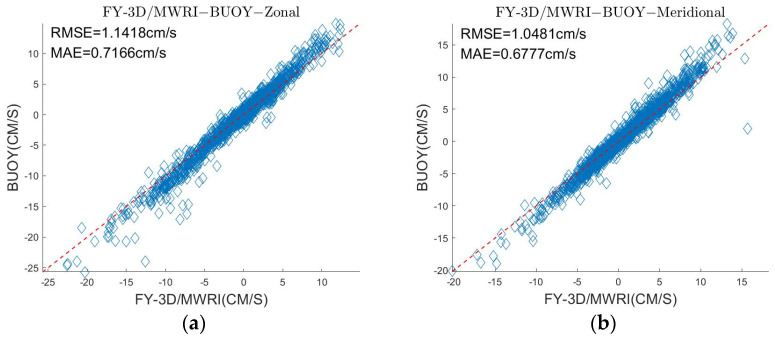
(**a**) is a comparison of the sea ice motion obtained by using FY-3D/MWRI T*_b_* data and the motion of the buoy in the zonal direction. (**b**) is a comparison of the sea ice motion retrieved by using FY-3D/MWRI T*_b_* data and the motion of the buoy in the meridional direction. (**c**) is a comparison of the sea ice motion obtained from the NSIDC and the motion of the buoy in the zonal direction. (**d**) is a comparison of the sea ice motion obtained from the NSIDC and the motion of the buoy in the meridional direction.

**Figure 12 sensors-22-08298-f012:**
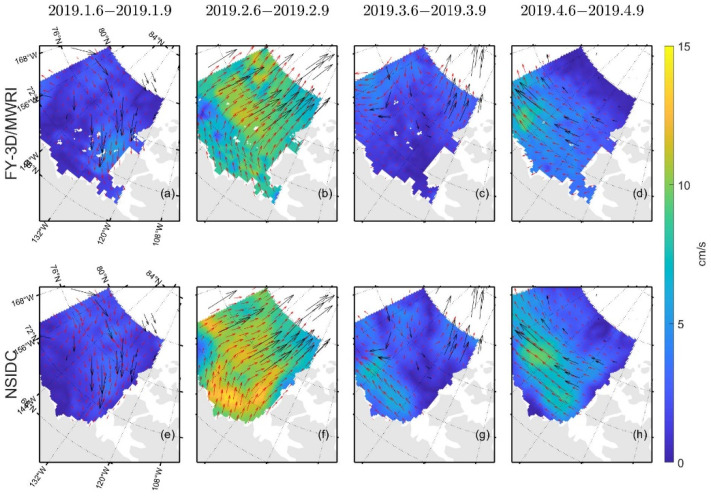
Schematic diagram of the sea ice motion retrieved by using FY-3D/MWRI T*_b_* data and the corresponding NSIDC product. Figure (**a**–**d**) show sea ice motion retrieved from FY-3D/MWRI T*_b_* data. Figure (**e**–**h**) show sea ice motion provided by NSIDC.

**Figure 13 sensors-22-08298-f013:**
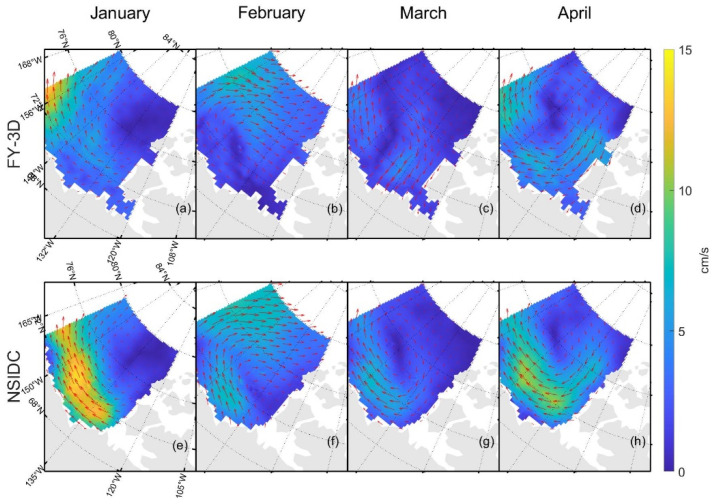
Schematic diagram of the monthly average product for the sea ice motion. Figure (**a**–**d**) show sea ice motion retrieved from FY-3D/MWRI T*_b_* data. Figure (**e**–**h**) show sea ice motion provided by NSIDC.

**Figure 14 sensors-22-08298-f014:**
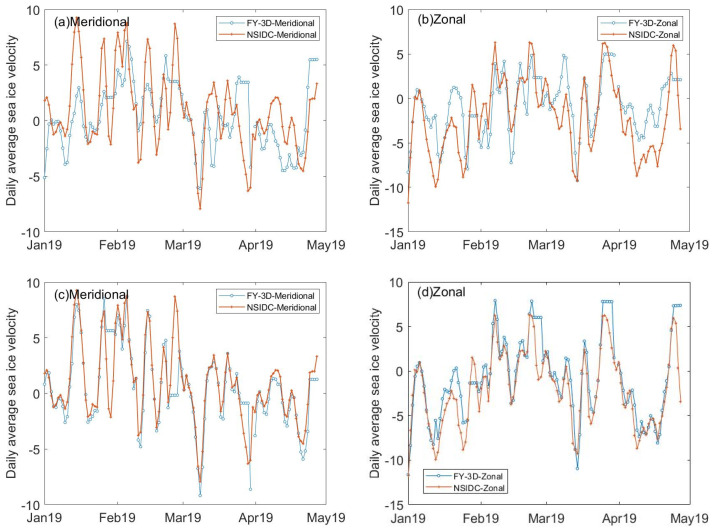
Comparisons of daily average velocities. (**a**,**b**) are the comparisons of the sea ice motion obtained from the NSDIC product and retrieved from FY-3D/MWRI T*_b_* data (combining the different polarizations and frequencies) in the meridional and zonal direction, respectively. (**c**,**d**) are the comparisons of sea ice motion obtained from the NSIDC product and retrieved from FY-3D/MWRI T*_b_* data with data assimilation in the meridional and zonal direction, respectively.

**Table 1 sensors-22-08298-t001:** Basic information of the data used in this study.

Data	Source	Spatial Resolution
FY-3D/MWRI T*_b_* data	NSMC	9 km × 15 km (89 GHz)
		18 km × 30 km (36.5 GHz)
Sea ice concentration data	POGOC	12.5 km × 12.5 km
Buoy data	IABP	
Sea ice motion data product	NSIDC	25 km × 25 km
NCEP/NCAR wind data	NCEP/NCAR	about 100 km × 100 km ^1^

^1.^ NCEP–NCAR winds are on a T62 Gaussian grid, which is ∼100 km in the latitudinal direction, with variable longitudinal spacing [14].

**Table 2 sensors-22-08298-t002:** Information on the parameters of FY-3D/MWRI [30].

Frequency (GHz)	Spatial Resolution (km)	Bandwidth (MHz)	Sensitivity (K)	Polarization *
36.5	18 × 30	900	0.5	H/V
89	9 × 15	2 × 2300	0.8	H/V

* We used only data at frequencies of 36.5 GHz and 89 GHz in this study. H represents the horizontal polarization and V represents the vertical polarization.

**Table 3 sensors-22-08298-t003:** Error-related statistics of the retrieved sea ice motion from FY-3D/MWRI T*_b_* data at 89 GHz in the Beaufort Sea.

	H		V	
	Zonal δ/ σ *	Meridional δ/ σ	Zonal δ/ σ	Meridional δ/ σ
January	2.4266 (3.5419)	2.9908 (4.4169)	2.3972 (3.5194)	2.9971 (4.4061)
February	3.3967 (4.0409)	2.9113 (3.7086)	3.4036 (4.0444)	2.8997 (3.6931)
March	1.9316 (2.6656)	2.9561 (4.2914)	1.9372 (2.6595)	2.9451 (4.2475)
April	3.3281 (4.3451)	4.0764 (5.2480)	3.3317 (4.3552)	3.9544 (5.1051)
January–April	2.7472 (3.6763)	3.2017 (4.4202)	2.7392 (3.6695)	3.1658 (4.3615)

* In each set of numbers, the first represents δ, the number in parentheses represents σ, and the unit is cm/s.

**Table 4 sensors-22-08298-t004:** Error-related statistics of the retrieved sea ice motion from FY-3D/MWRI T*_b_* data at 36.5 GHz in the Beaufort Sea.

	H		V	
	Zonal δ/ σ *	Meridional δ/ σ	Zonal δ/ σ	Meridional δ/ σ
January	2.2732 (3.0348)	3.0680 (4.4169)	2.2668 (3.0320)	3.1237 (4.6500)
February	3.3573 (4.0050)	2.9241 (3.7044)	3.4065 (4.0683)	2.9804 (3.7950)
March	1.8568 (2.5713)	2.9275 (4.1990)	1.8389 (2.5900)	2.8325 (4.0852)
April	3.0200 (3.9662)	3.8212 (4.9949)	3.1476 (4.0604)	4.0861 (5.3117)
January–April	2.5793 (3.3887)	3.1273 (4.3345)	2.6166 (3.4415)	3.1842 (4.4247)

* In each set of numbers, the first represents δ, the number in parentheses represents σ, and the unit is cm/s.

**Table 5 sensors-22-08298-t005:** Error-related statistics of the retrieved sea ice motion by using FY-3D/MWRI after merging the results of different frequencies and polarizations.

	Zonal δ/ σ *	Meridional δ/ σ
January	2.3136 (3.4180)	3.1575 (4.7102)
February	3.3406 (3.9833)	2.9441 (3.7419)
March	1.8334 (2.5688)	2.8484 (4.1092)
April	3.3483 (4.3538)	4.1328 (5.3143)
January–April	2.6824 (3.6159)	3.2437 (4.4857)

* In each set of numbers, the first represents the δ, and the number in parenthese represents the σ, the unit is cm/s.

**Table 6 sensors-22-08298-t006:** Error-related statistics of the retrieved sea ice motion from FY-3D/MWRI T*_b_* and NSIDC.

	FY-3D/MWRI		NSIDC	
	Zonal δ/ σ *	Meridional δ/ σ	Zonal δ/ σ	Meridional δ/ σ
January	0.6226 (1.2170)	0.7552 (1.3285)	0.6035 (0.8863)	0.4560 (0.6744)
February	0.7683 (1.0312)	0.6352 (0.8305)	0.5201 (0.6791)	0.6184 (0.7631)
March	0.5991 (0.9793)	0.5820 (0.8219)	0.5742 (0.7885)	0.4567 (0.6057)
April	0.9163 (1.3123)	0.7335 (1.0797)	0.6630 (0.8909)	0.4648 (0.6383)
January–April	0.7166 (1.1418)	0.6777 (1.0481)	0.6576 (0.9515)	0.4922 (0.6700)

* In each set of numbers, the first is δ, the number in parenthesis is σ, and the unit is cm/s.

## Data Availability

FY-3D/MWRI brightness temperature data were provided by the NSMC (http://www.nsmc.org.cn/) accessed on 26 August 2020. Sea ice motion product was provided by the NSIDC (https://nsidc.org/data/nsidc-0116/versions/4/) accessed on 26 August 2020. The wind velocity data were provided by NCEP/NCAR (https://psl.noaa.gov/data/gridded/data.ncep.reanalysis.html) accessed 10 April 2021. The buoy data was provided by IABP (https://iabp.apl.uw.edu/) accessed on 26 August 2020. The Sea Ice Concentration data were released in the POGOC website (http://coas.ouc.edu.cn/pogoc/sy/list.htm) accessed on 22 April 2021.

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
