# Peer review of "Retrieving the Motion of Beaufort Sea Ice Using Brightness Temperature Data from FY-3D Microwave Radiometer Imager"

_sensors, 2022, doi:10.3390/s22218298_

Round 1

Reviewer 1 Report

This manuscript presents the ice velocity result according to the established maximum cross-correlation (MCC) and the successive correction method (SCM), using the brightness temperature data from FY-3D. The idea and structure of the manuscript are ok to me. However, there are too many figures (17) and some of them are similar. And corresponding descriptions and analysis on these figures are not enough. More worrying is that part result is circular reasoning. That is, there is no mathematical meaning if comparing x with y determined by exactly the same x.

The following are some specific comments.

Abstract

L12 There are no results about the weights in the main text.

Method

L192 wrong characters.

What is the full name of IDW?

Results

L234-250 This part belongs to the method section

L251-341 There are too many figures (up to 9) in this section. Whereas, the corresponding descriptions and analysis are missing.

Don’t use RMSE and its abbreviation σ at the same time.

Wrong capitalization format in Figure 6

Figure 6-9 I don’t understand why respectively display the ice velocity in zonal and meridional direction here. But not the component velocity.

L343-355 This part belongs to the method.

L350 In the previous part (L130), “we merged the data on wind velocity with those on the velocity of sea ice at a rate of 1%”. But introduce a weighted method here. Furthermore, the descriptions of Eq. 10 are not clear.

L368-378 This part is a kind of circular reasoning. There is no meaning in the comparing result (fig 14).

Fig 15 and 16 There are not any mentions for both figures. What is the meaning of them?

L399 Wrong figure numbers.

Discussion

L421-428 circular reasoning

Conclusion

The RMSE from circular reasoning is still larger than the RMSE of NSIDC data. 

Reviewer 2 Report

I have gone through the manuscript with titled as “Retrieving the Motion of Beaufort Sea Ice Using Brightness Temperature Data from FY-3D Microwave Radiometer Imager”.

This manuscript designed the MCC and SCM to retrieve the motion of sea ice in Beaufort Sea, and used the Buoy and NSIDC products to validate the proposed method. The comparisons show significant results, and the overall work is interesting. However, some issues need to be addressed before reconsidering publication.

1) Abstract: The background or motivation of this manuscript is lack, and the details of the method based on MCC and SCM are unclear.

2) Introduction: the motivation of this manuscript is not clear or lack, please address this issue.

3) Data and Method-Section 2.3. In my opinion, this section includes two key issues, one is the the improved sea ice motion method based on MCC, and the other is SCM. However, the SCM is not addressed, and the data preprocessing is contained in Figure 1. Also, 2.3.2LOG filter is a key step? Or a step of data preprocessing. Similarly, what is function of the data assimilation? It relates with SCM? What is the core idea of the SCM.

4)Discussion: The products of the motion of sea ice obtained by the improved MCC method are compared with those from Buoy data. As addressed in 2.3.3, two-thirds of buoy data are used for assimilation and one-third is used to test the results of data assimilation. Thus, there is no doubt that the comparisons between the proposed products and the buoy data is well agreed.

The minor revisions

1)Figure 1 needs to be redrawn

2) The used language is not always fluent apart some typos spread over the manuscript

Reviewer 3 Report

The manuscript presented the retrieval analysis of sea ice motion using data of Microwave Radiometer Imager onboard Fengyun-3D satellite and other data sources. This paper has serious flaws, and major comments are summarized as follows

(1)      The content of this paper is not rigorous. Some of the presented results and related discussion and conclusion are inconsistent. The contribution or novelty of this work is unclear based on the presented results. Many grammar mistakes and inadequate description make the paper very hard to read and understand, and also very hard to evaluate the correctness of the results. The structure of some sections of the paper is disordered, repeated and confusing.

(2)      Summary of the existing research work in the Introduction is not clear, unreadable, and not sufficient, and great improvement is needed.

(3)      Fengyun-3 is a series of meteorological satellites, which includes Fengyun-3A, Fengyun-3B, etc. It is totally wrong that the author presented ” The Fengyun-3 (FY-3) meteorological satellite is China's second generation of polar-orbiting meteorological satellite that was launched in 2017” in the Introduction.

(4)      What do you mean by ” The relevant studies in the Arctic region have focused on calculating sea ice concentration and thickness of snow based on evaluations of the data on Tb [26-29].”? It’s hard to understand what the author what to present when you read this sentence as a part of that paragraph.

(5)      There is no data shown in Table 1. Please correct your description in page 3, line 97 ” The data are shown in the Table 1.” There are so many sentences that have similar problem, which make the whole manuscript hard to understand. In addition, some information are wrong in Table 1.

(6)      What is ”FY-3D/MWRI data on Tb”. It’s not a common description of MWRI data, and the readers can not understand what data you are using. Same problem can be observed for other datasets that the authors claimed that they are using.

(7)      Why two sensing frequencies are selected for this study?

(8)      Why SIC data used is retrieved from MWRI onboard FY-3B instead of FY-3D?

(9)      The time should be given with time zone for every dataset you used.

(10)    Why day 1 and day 4 is considered to calculate the three-day average velocity of the buoy?

(11)    What do you mean by the word ” merge” in page 4, line 131? What is the detailed procedure for this?

(12)    What’s the purpose for the inter-calibration between MWRI and AMSR-2? How did you perform the inter-calibration? Comparison results should be discussed with and without the inter-calibration.

(13)    The data description and data preprocessing procedure sections are very confusing. These vital basic information should be described clearly for readers to understand.

(14)    Why only day 1 and day 4 data are analyzed and day 2 data is ignored in this study?

(15)    What do you mean by ” a template of size 7 x 7”?

(16)    What is the selection criteria for the threshold of the cross-correlation coefficient?

(17)    The description should be given more detailed for page 4, line164 to 168. The current version can not be understood.

(18)    What is IDM as given in Figure 1?

(19)    What is IDW method?

(20)    The procedure in Figure 1 is totally different compared with the description for it.

(21)    What do you mean by m x n, no sign of n in Figure 2.

(22)    Section 2.3.1, section 2.3.2 and section 2.3.3 only presented some basic equations of the method, but lack of the discussion of the usage to this study.

(23)    Authors claimed that ” Tables 3 and 4 show the error in the retrieved motion of sea ice”, what is the error?

(24)    It is hard to differentiate the red and black arrows in figures 4,5,12,15. How do you define the 7x7 grids?

(25)    How did you get figures 6 and 7, what is the time period and spatial area? The basic description must be given.

(26)    How did you get the zonal and meridional results separately?

(27)    What’s the relationship between the buoy and retrieved sea ice velocity? What causes the difference between them? What’s the meaning for presenting and analyzing figures 6 to 9?

(28)    What’s the calculation method for each monthly based result in tables 3 and 4?

(29)    How do you define an error is large or not? How can you get the conclusion in page 11, line 300-line302?

(30)    What’s the physical principle for equations 8 and 9?

(31)    No improvement can be observed when comparing figures 11 and 12 with figures 6 to 9. What’s meaning for the analysis in section 3.1?

(32)    Spelling mistakes in the title of section 3.2.

(33)    What’s the difference between section 2.3.3 and section 3.2?

(34)    The authors did comparison between some products with different spatial resolution and unknown time resolution. The comparison method should be discussed in detail to make the comparison make sense.

(35)    The IABP and NSIDC data are with highly consistency, but the FY-3D results have many unique values as shown in Figure 17. The reason for these unique results should be discussed.

(36)    Authors concluded in the Discussion section that” The above comparisons of the motion of sea ice retrieved by using the improved MCC method, show that the quality of the data improved considerably by merging the data.”, which is unreasonable and meaningless. Because better results can be obtained when only using the NSIDC data compared to the merged data.

(37)    Authors concluded that ” The comparison shows that the SCM significantly improve the  quality of the product for the motion of sea ice. ”, which is hard to understand. No comparison results are given for this kind of comparison.

(38)    The last two paragraph of the discussion section is impossible to be understood for the readers since no related results have been ever presented in the results section.

(39)    Tb needs to be defined when you use it for the first time. (page 2, line 45). There are some similar mistakes need to be addressed.

(40)    Spelling mistakes like Quick Scaterometer and so on need to be corrected.

Round 2

Reviewer 1 Report

I read throughout the manuscript and felt that the authors have addressed all my issues. I have no more comments on the manuswcript. The current style is suitable to be accepted by the journal.

Author Response

Thank you for your comments.

Reviewer 2 Report

Thanks for the authors' responses.

Most of the lastset comments have been addressed, and some references have been appended. However, the motivation of this mauscript is still unclear. Exception of the motivation, in my opinion, this revised manuscript may be acceptable.

Reviewer 3 Report

The current version looks good.